# Assessment of Stability and Adaptation Patterns of White Sugar Yield from Sugar Beet Cultivars in Temperate Climate Environments

**Marcin Studnicki** [1,*] **, Tomasz Lenartowicz** [2]**, Kinga Noras** [1] **, Elżbieta Wójcik-Gront** [1] **and Zdzisław Wyszyński** [3]

1   Department of Experimental Design and Bioinformatics, Warsaw University of Life Sciences—SGGW, 02-766 Warsaw, Poland
2   Research Center for Cultivar Testing (COBORU), 63-022 Słupia Wielka, Poland
3   Department of Agronomy, Warsaw University of Life Science—GGW, 02-766 Warsaw, Poland
*   Correspondence: marcin_studnicki@sggw.pl; Tel.:+4-22-593-2727

**Abstract:** The yield and yield quality of sugar from the sugar beet (*Beta vulgaris* L.) and are determined by genotype, environment and crop management. This study was aimed at analyzing the stability of white sugar yield and the adaptation of cultivars based on 36 modern sugar beet cultivars under different environmental conditions. The compatibility of sugar beet cultivars' rankings between the three growing seasons and between the 11 examined locations was assessed. In addition, an attempt was made to group environments to create mega-environments. From among the 11 examined locations, four mega-environments were distinguished on the basis of the compatibility of the white sugar yield rankings. The assessment of the adaptation of cultivars and the determination of mega-environments was carried out using GGE (genotype main effects plus genotype environment interaction effects) biplots and confirmed by the Spearman rank correlation test performed for cultivars between locations. The cultivars studied were characterized by a high stability of white sugar yield in the considered growing seasons. The high compliance of the sugar yield rankings between the years contributes to a more effective recommendation of cultivars.

**Keywords:** genotype × environment interaction; mega-environments; sugar beet; white sugar yield

## 1. Introduction

The sugar beet (*Beta vulgaris* L.) is one of the most important plants used in the processing industry in Europe. The vast majority of sugar produced in Europe comes from this plant. The area where the sugar beet was cultivated in 2017 was over 4.8 million ha in the world, of which 3.3 million ha were in Europe [1,2]. Beet growing is becoming more and more important in the production of biofuels [3,4]. Officially, in Europe in the Common Catalogue of Varieties of Agricultural Plant Species (CCA list) over 1500 sugar beet cultivars were registered. Thus, a proper recommendation will allow farmers to choose cultivars appropriate to the environmental conditions of their fields.

The recommendations are based on the assessment of the adaptation abilities of cultivars to an individual environment or group of environments, also known as an agro–ecological region or mega-environment. It is also important to assess the stability of the cultivar in accordance with the dynamic concept [5], which defines a stable cultivar as the one that has a yield parallel to the average yield of all cultivars (environmental means) in the studied environments. The adaptive patterns of cultivars and their stability are related to the strength of the genotype–environment interactions (G × E). The most stable cultivars are characterized by a negligible effect of genotype by environment

interaction. This is especially important for an industry that is looking for the best possible quality and a stable amount of raw material [6,7]. In the sugar production, the most important quality, in addition to the yield of roots, is the sugar content and, consequently, the sugar yield obtained by a given cultivar. Usually, the evaluation of the G × E interactions and patterns of adaptation of cultivars takes place using multi environmental trails (METs) [8–10].

In the world, there are many cultivar evaluation systems used for giving proper recommendations for farmers, e.g., the Potato National Chip Processing Trial [6] and the Iowa Crop Performance Test for soybean and corn [11]. An exemplary system for the sugar beet is the Polish Post-Registration Variety Testing System (PVTS). The PVTS was established in 1998 for major field crops (cereals, potatoes and sugar beets) by the Research Centre of Cultivar Testing (COBORU), and it works in coordination with the COBORU with contributions and support from regional administrative and agricultural bodies, breeding and seed companies, extension-service agencies and farmers' associations (http://www.coboru.pl).

So far, the variability assessment of cultivars for many different types of environments has been widely carried out for cereals [8,12]. However, there is no detailed research on the adaptive patterns of modern sugar beet cultivars. Thus, the aim of this research was (I) to assess the stability of the white sugar yield and adaptation of 36 modern sugar beet cultivars in temperate environmental conditions; (II) to assess the compatibility of sugar yield rankings between studied growing seasons and locations; (III) to attempt to combine similar environments in order to create mega-environments.

## 2. Materials and Methods

### 2.1. Field Trials

The data on white sugar yield used in this study were obtained from 11 locations (L) (Table 1) of the Polish Post-Registration Variety Testing System (PVTS) for 36 sugar beet cultivars (G) (Table 2) during three growing seasons (Y) from 2015 until 2017. About 20 cultivars were studied during each growing season; therefore, the G × L × Y data set (36 cultivars, 11 locations and 3 seasons) was unbalanced. Each field experiment was carried out in accordance with a complete block design with four replications. The crop management in the trials included standard fertilization adapted to conditions in each location, the interventional use of herbicides and insecticides, and seed treatment. The plot area was 11 m$^2$. After the harvest, root yield was determined. Quality traits such as sugar content, potassium, sodium and $\alpha$-amino-nitrogen were determined using Venema automatic laboratory systems. White sugar yield was obtained from root yield and quality parameters used in the standard formula [13].

**Table 1.** Description of 11 trial locations in the Polish Post-Registration Variety Testing System within three growing seasons.

| Location | Latitude Longitude | Growing Seasons | Soil Texture [a] | Soil Classification | Soil Fertility and Climate Conditions Category [b] | pH |
|---|---|---|---|---|---|---|
| Bezek | 51º12′6.384″ N 23º16′7.795″ E | 2015 | CL | Cambisols | 2 | 7.35 |
| | | 2016 | CL | Cambisols | 3 | 7.4 |
| | | 2017 | CL | Cambisols | 2 | 7.26 |
| Chrzastowo | 53º10′0″ N 17º34′60″ E | 2015 | SL | Luvisols | 2 | 6.5 |
| | | 2016 | SL | Luvisols | 2 | 5.9 |
| | | 2017 | SL | Luvisols | 2 | 6.3 |
| Czeslawice | 51º18′21.969″ N 22º15′55.652″ E | 2016 | L | Luvisols | 2 | 6.4 |
| | | 2017 | L | Luvisols | 2 | 6.2 |
| Glebokie | 52º36′46.014″ N 18º26′47.286″ E | 2015 | SL | Phaeozems | 2 | 6.9 |
| | | 2016 | SL | Phaeozems | 2 | 6.9 |

**Table 1.** *Cont.*

| Location | Latitude Longitude | Growing Seasons | Soil Texture [a] | Soil Classification | Soil Fertility and Climate Conditions Category [b] | pH |
|---|---|---|---|---|---|---|
| Glubczyce | 50°12′0.339″ N 17°49′45.552″ E | 2015 | L | Luvisols | 1 | 6.7 |
| | | 2016 | L | Luvisols | 1 | 6.6 |
| | | 2017 | L | Luvisols | 1 | 6.5 |
| Kaweczyn | 51°56′27.183″ N 20°54′23.297″ E | 2015 | SL | Luvisols | 4 | 6.47 |
| | | 2016 | SL | Luvisols | 4 | 6.39 |
| | | 2017 | SL | Luvisols | 4 | 6.3 |
| KoscielnaWies | 51°47′5.772″ N 18°0′36.535″ E | 2015 | SL | Cambisols | 2 | 6.7 |
| | | 2016 | SL | Cambisols | 2 | 6.3 |
| | | 2017 | SL | Cambisols | 2 | 6.3 |
| Lisewo | 53°58′17.701″ N 18°6′35.516″ E | 2015 | SiL | Fluvisols | 1 | 5.9 |
| | | 2016 | SiL | Fluvisols | 1 | 5.9 |
| | | 2017 | SiL | Fluvisols | 1 | 6.7 |
| Przeclaw | 50°11′38.107″ N 21°28′47.483″ E | 2016 | LS | Fluvisols | 1 | 7.38 |
| | | 2017 | LS | Fluvisols | 2 | 7.2 |
| SlupiaWielka | 52°13′4.759″ N 17°13′6.241″ E | 2015 | CL | Phaeozems | 2 | 6.4 |
| | | 2016 | LS | Phaeozems | 1 | 6.3 |
| | | 2017 | SL | Phaeozems | 2 | 7.4 |
| Zybiszow | 51°3′50.332″ N 16°54′41.103″ E | 2016 | LS | Chernozems | 2 | 6.3 |
| | | 2017 | LS | Chernozems | 1 | 6.5 |

[a] SL: Sandy loam; CL: Clay loam; L: Loam; SiL: Silt loam; LS: Loamy sand; [b] The Polish system of evaluating soil fertility and climate conditions e from 1 (best) to 9 (worst).

**Table 2.** Characteristics of the studied 36 cultivars and the basic statistics of white sugar yield t ha$^{-1}$across the years and trial locations.

| Cultivar | Year of Registrationu | Breeders | Country of Origin | White Sugar Yield Across Study Location | | | |
|---|---|---|---|---|---|---|---|
| | | | | Min | Max | Mean | SD |
| Amazonia | 2015 | SESVANDERHAVE N.V./S.A. | BE | 7.93 | 18.78 | 14.23 | 3.09 |
| Bravo | 2015 | MariboHilleshög ApS | DK | 8.40 | 18.85 | 14.20 | 2.98 |
| Bravura | 2017 | DLF Seeds A/S | DK | 8.11 | 18.77 | 14.22 | 3.02 |
| BTS 2160 | 2016 | Friedrich-Ebert-Anlage 36 | DE | 7.94 | 18.69 | 14.27 | 3.01 |
| BTS 350 | 2016 | Friedrich-Ebert-Anlage 36 | DE | 6.80 | 19.52 | 14.50 | 3.48 |
| Candimax | 2017 | SAS Florimond Desprez Veuve & Fils | FR | 7.86 | 19.17 | 14.15 | 3.24 |
| Contenta | 2015 | DLF Seeds A/S | DK | 8.06 | 18.16 | 13.60 | 2.81 |
| Denzel | 2015 | Strube Research GmbH & Co. KG | DE | 8.27 | 18.73 | 14.15 | 3.04 |
| Diplomat | 2017 | MariboHilleshög ApS | DK | 7.64 | 19.35 | 14.08 | 3.34 |
| Doppler | 2016 | Strube Research GmbH & Co. KG | DE | 7.85 | 19.52 | 14.02 | 3.26 |
| Exotique | 2017 | SAS Florimond Desprez Veuve & Fils | FR | 7.63 | 18.88 | 14.36 | 3.20 |
| Fala | 2016 | SESVANDERHAVE N.V./S.A. | BE | 7.79 | 19.22 | 14.10 | 3.31 |
| Fantazja | 2015 | Kutnowska Hodowla Buraka Cukrowego sp. z o.o. | PL | 8.82 | 18.84 | 14.01 | 3.08 |
| FD Taekwondo | 2016 | SAS Florimond Desprez Veuve & Fils | FR | 7.79 | 18.37 | 14.26 | 2.94 |
| Hammond | 2015 | Strube Research GmbH & Co. KG | DE | 7.61 | 18.89 | 14.15 | 3.15 |
| Igloo | 2014 | SESVANDERHAVE N.V./S.A. | BE | 7.27 | 18.89 | 14.36 | 3.28 |
| Jadeit | 2016 | Kutnowska Hodowla Buraka Cukrowego sp. z o.o. | PL | 7.75 | 19.26 | 14.38 | 3.21 |
| Jagger | 2015 | Strube Research GmbH & Co. KG | DE | 7.35 | 19.02 | 14.14 | 3.30 |

**Table 2.** *Cont.*

| Cultivar | Year of Registrationu | Breeders | Country of Origin | White Sugar Yield Across Study Location | | | |
|---|---|---|---|---|---|---|---|
| | | | | Min | Max | Mean | SD |
| Kagu | 2016 | SESVANDERHAVE N.V./S.A. | BE | 8.32 | 18.87 | 14.24 | 3.03 |
| Krajan | 2016 | Kutnowska Hodowla Buraka Cukrowego sp. z o.o. | PL | 7.81 | 19.33 | 14.20 | 3.13 |
| Kujavia | 2017 | Kutnowska Hodowla Buraka Cukrowego sp. z o.o. | PL | 7.76 | 19.73 | 14.44 | 3.38 |
| Marenka KWS | 2016 | Grimsehl Strasse 31 | DE | 7.49 | 18.45 | 14.08 | 2.98 |
| Marinus | 2015 | Strube Research GmbH & Co. KG | DE | 8.79 | 18.93 | 13.97 | 3.11 |
| Marynia | 2017 | Wielkopolska Hodowla Buraka Cukrowego sp. z o.o. | PL | 8.70 | 19.05 | 13.99 | 3.15 |
| Mazur | 2017 | Strube Research GmbH & Co. KG | DE | 8.15 | 19.12 | 14.35 | 3.19 |
| Mélusine | 2017 | SAS Florimond Desprez Veuve & Fils | FR | 7.73 | 19.19 | 14.20 | 3.27 |
| Mesange | 2014 | SAS Florimond Desprez Veuve & Fils | FR | 8.00 | 18.93 | 14.14 | 3.12 |
| Panorama KWS | 2014 | Wielkopolska Hodowla Buraka Cukrowego sp. z o.o. | PL | 7.59 | 19.26 | 14.69 | 3.14 |
| Polanin | 2016 | Wielkopolska Hodowla Buraka Cukrowego sp. z o.o. | PL | 8.80 | 19.57 | 14.05 | 3.11 |
| Polmar | 2016 | MariboHilleshög ApS | DK | 8.34 | 18.63 | 14.11 | 3.01 |
| Silezja | 2015 | Kutnowska Hodowla Buraka Cukrowego sp. z o.o. | PL | 7.44 | 19.30 | 14.41 | 3.25 |
| Sobieski | 2015 | Wielkopolska Hodowla Buraka Cukrowego sp. z o.o. | PL | 8.77 | 18.79 | 14.15 | 3.03 |
| Sombrero | 2017 | SESVANDERHAVE N.V./S.A. | BE | 8.09 | 18.75 | 14.19 | 3.06 |
| Sukcesja KWS | 2014 | KWS Saat SE | DE | 7.70 | 18.78 | 14.35 | 3.10 |
| Tapir | 2014 | SESVANDERHAVE N.V./S.A. | BE | 7.54 | 19.29 | 14.16 | 3.35 |
| Toleranza KWS | 2015 | KWS Saat SE | DE | 7.59 | 19.17 | 14.68 | 3.12 |

## 2.2. Statistical Methods

The analysis of white sugar yield data was performed using a single-stage approach with a linear mixed model (LMM). The LMM used for the complete block design is given by the equation:

$$y_{ijhkl} = \mu + l_j + g_k + a_i + gl_{kj} + ga_{ki} + la_{ji} + gla_{kji} + b_{jih} + e_{ijhkl} \tag{1}$$

where $\mu$ is the overall mean; $l_j$ is the fixed effect of the $j$-th location; $g_k$ is the random effect of the $k$-th cultivar; $a_i$ is the random effect of the $i$-th year; $gl_{kj}$ is the random interaction effect of the $k$-th cultivar and $j$-th location; $ga_{ki}$ is the random interaction effect of the $k$-th cultivar and $i$-th year; $la_{ji}$ is the random interaction effect of the $j$-th location and i-th year; $gla_{kji}$ is the random interaction effect of the $k$-th cultivar, $j$-th location, and $i$-th year; $b_{jih}$ is the random effect of the $h$-th block nested in $j$-th location at $i$-th year; and $e_{ijhkl}$ is the random effect of error associated with the white sugar yield observation $y_{ijhkl}$.

The cultivar random effects in each location were modeled using a factor analytic (FA) structure with two components. The FA structure used multiplicative terms to approximate the unstructured variance–covariance matrix. Two tests were used to estimate the significance of the main and interaction effects in the presented LMM—the Wald F test was used for fixed effects, and variance components were used for random effects. In the LMM, the adjusted means of white sugar yield for combinations cultivar × location and cultivar × location × year were calculated using the algorithm described by [14], obtained on the basis of BLUP (best linear unbiased prediction) for random effects and BLUE (best linear unbiased estimator) for fixed effects. The variance parameters and BLUP were estimated using the restricted maximum likelihood (REML) method. REML estimation methods have been commonly used in the case of unbalanced data observed in multi-environmental trials [15,16]. The likelihood ratio test was used to evaluate the significance of the variance components.

The obtained adjusted means of white sugar yield for relevant combinations were used to assess the compatibility of cultivar rankings in the studied agro–ecological regions, growing seasons and trial locations. This assessment was performed to evaluate the repeatability of the white sugar yield of a given cultivar in years and in different trial locations in each agro–ecological region. The compatibility of cultivars rankings for white sugar yield in different years and trial locations was evaluated using the Spearman rank correlation coefficient. A value of the coefficient higher than 0.7 indicated the compatibility of the white sugar yield rankings in regions or trial locations [17]. Additionally, we used a GGE (genotype main effects plus genotype environment interaction effects) biplot analysis based on the adjusted means of white sugar yield for cultivar × trial location [18].

For the statistical analysis, we used the R 3.5.1 software package. The applied LMM was fitted using ASReml 4.0 and implemented in the R software package ASReml-R 3 [19]. The GGE biplot analysis was performed using the gge 1.4 package [20].

## 3. Results

Among the considered random effects, the greatest degree of total white sugar yield variation was explained by the interaction between the year and the location −77% (Table 3). The yield of sugar was also influenced by the random effect of the year −19%. The effect of cultivar only explained approximately 0.5% of the total variation of the white sugar yield. The only fixed main effect, location, was significant.

**Table 3.** The Wald F ratio for fixed effects and variance components with their percentage share in the total variation of random effects for the white sugar yield.

| | **Fixed Effects** | | |
|---|---|---|---|
| **Study Effect** | **Wald F** | | ***p* Value** |
| Location | 11.496 | | 0.00371 |
| | Random effects | | |
| Study effect | Variance Components | *p* value | Percent of total variance |
| Year | 1.952 | 0.0018 | 18.84 |
| Cultivar | 0.040* | 0.0351 | 0.39 |
| Year * Location | 7.995** | 0.0003 | 77.19 |
| Year * Cultivar | 0.018 | 0.1841 | 0.17 |
| Cultivar * Location | 0.177* | 0.0114 | 1.71 |
| Year * Cultivar * Location | 0.176* | 0.0124 | 1.70 |

The means of white sugar yield from sugar beet cultivars ranged from 13.60 t ha$^{-1}$ for the Contenta cultivar to 14.69 t ha$^{-1}$ for the Panorama KWS cultivar (Table 2). The difference between the best-yielding cultivar and the weakest one was not large, amounting to just over 1 ton. The variation in the white sugar yield for the tested cultivars was also very similar, ranging from around 8 to 18 t ha$^{-1}$, and the average standard deviation was 3.15 t ha$^{-1}$. The means of white sugar yield in trial locations ranged from 8 t ha$^{-1}$ in Koscielna Wies to 17.32 t ha$^{-1}$ in Zybiszow (Table 4). The variation in white sugar yield for cultivars in trial locations was very high, e.g., the standard deviation ranked from 0.46 t ha$^{-1}$ in Koscielna Wies to 3.32 t ha$^{-1}$ in Bezek.

Based on the GGE biplot analysis (Figure 1), we identified four groups of trial locations with similar adaptations of cultivars with respect to the white sugar yield. In the first group, there was only one location, Glebokie, where the following widely adapted cultivars were distinguished: Exotique, Panorama KWS, and Toleranza KWS. Group 2 included six locations—Brezek, Chrzastowo, Lisewo, Przeclaw, Slupia Wielka and Zybiszow. In these locations, cultivars BTS 350, Igloo and Silezja can be considered as widely adapted. The locations Glubczyce and Czeslawice fall into the next group of locations—group 3. Diplomat, Doppler, and Tapir cultivars were characterized by a wide adaptation in

this group of locations. The last group of locations (group 4) included Koscielna Wies and Kaweczyn. In this group of locations, cultivars Marinus, Polanin, and Sobieski can be considered as widely adapted.

**Table 4.** Basic statistics of white sugar yield (t ha$^{-1}$) for studied trial locations across studied years and cultivars.

| Trial Location | White Sugar Yield Across Study Studied Years and Cultivars | | | |
| --- | --- | --- | --- | --- |
| | Min | Max | Mean | SD |
| Bezek | 8.14 | 17.15 | 13.65 | 3.32 |
| Chrzastowo | 11.31 | 19.73 | 14.61 | 2.92 |
| Czeslawice | 10.37 | 18.25 | 15.04 | 2.15 |
| Glebokie | 10.22 | 14.79 | 12.06 | 1.09 |
| Glubczyce | 14.19 | 19.57 | 17.01 | 1.55 |
| Kaweczyn | 14.08 | 18.24 | 16.43 | 0.89 |
| Koscielna Wies | 6.80 | 8.95 | 8.00 | 0.46 |
| Lisewo | 8.77 | 15.27 | 12.48 | 2.12 |
| Przeclaw | 13.29 | 17.95 | 15.44 | 1.16 |
| SlupiaWielka | 10.96 | 18.19 | 14.29 | 1.68 |
| Zybiszow | 15.06 | 19.52 | 17.32 | 0.90 |

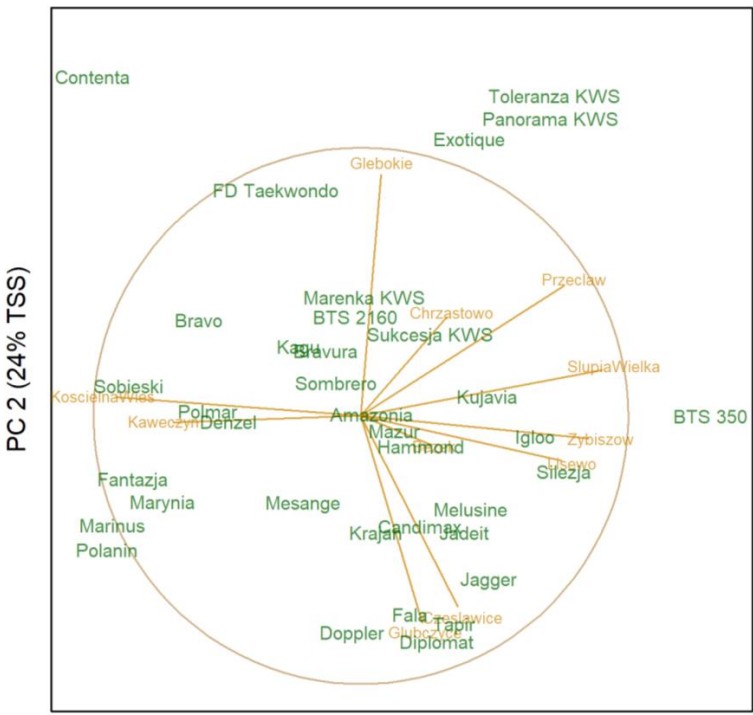

**Figure 1.** The GGE biplot based on adjusted means of the cultivar × trial location combinations for white sugar yield; PC = principal component; TSS = total sum of square.

The consistency of cultivar rankings for the white sugar yield between trial locations is presented in Figure 2. There was a strong positive correlation between locations within the groups identified above. The Spearman rank correlation coefficients were about 0.60 and even higher, which would indicate a high compatibility of the cultivar rankings. The strongest correlation was found between Koscielna Wies and Kaweczyn (group 4), with the coefficient equaling 0.87. We also identified places characterized by opposite ranking of cultivars, as evidenced by the negative correlation coefficient. The lowest value of the Spearman rank correlation coefficient was found between the rankings of cultivars in Kościelna Wieś and Słupia Wielka (−0.88). In general, an opposite ranking of cultivars was

observed between groups 4 2, which is presented in Figure 1—which contains the results of the GGA biplot analysis.

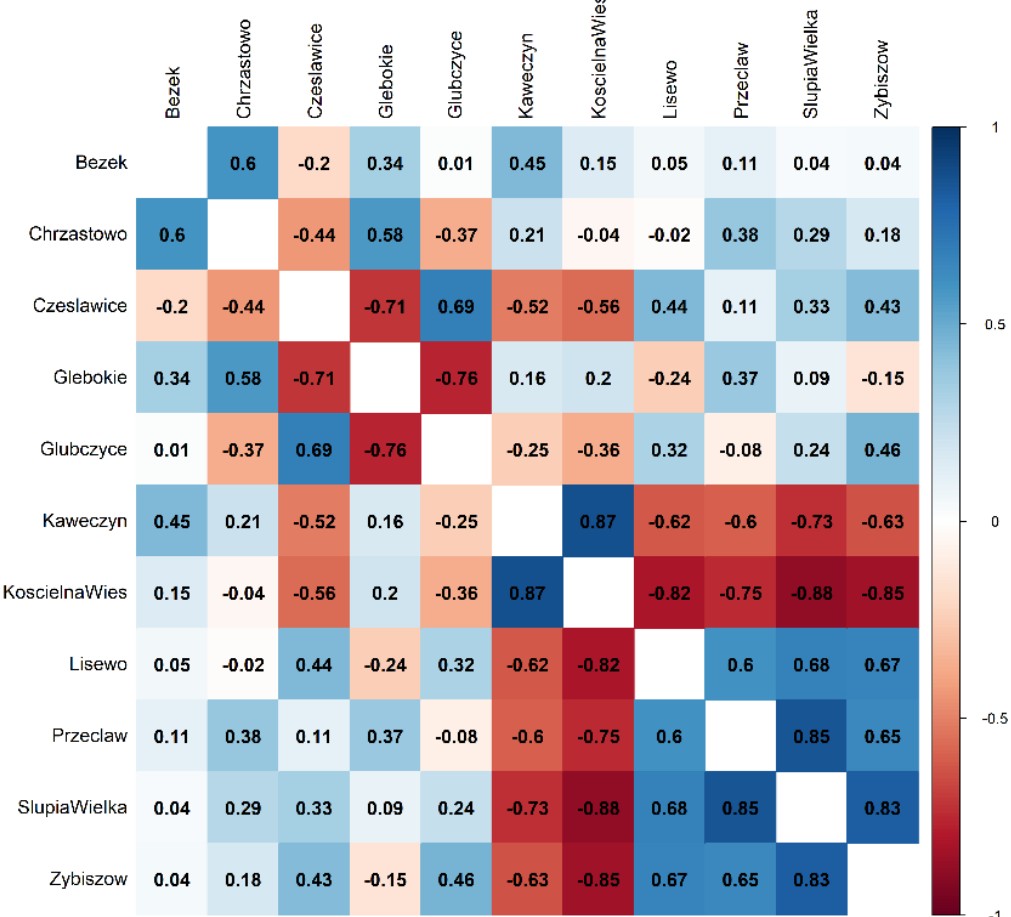

**Figure 2.** Spearman rank correlation coefficients of cultivars' white sugar yield between tested trial locations.

The correlation coefficients between the studied years were positive and above 0.70 (Figure 3). This proves the high compatibility of cultivars' rankings between all growing seasons.

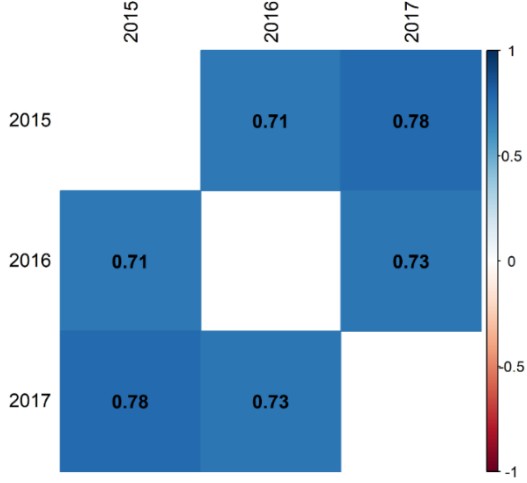

**Figure 3.** Spearman rank correlation coefficients of cultivars' white sugar yield between tested growing seasons.

## 4. Discussion

The assessment of adaptation carried out for sugar beet cultivars allowed us to indicate the cultivars which should be recommended for cultivation in a given region. It was not possible to indicate one cultivar or cultivars characterized by a wide adaptation to all environmental conditions in Poland and, thus, in other regions with a temperate climate. There was no cultivar with a stable and relatively high white sugar yield in all or in the vast majority of considered environments (trail locations) at the same time, i.e., a widely adapted cultivar. Sometimes, one or several cultivars with a wide adaptation to a large number of environments could be indicated, which is true for winter wheat cultivars in temperate climates [21,22]. However, for many agricultural plants, including the sugar beet, we could not identify cultivars with a wide adaptation [23,24]. Among the studied sugar beet cultivars, such a broad adaptation was not identified. As a result, it is not possible to recommend to farmers a universal cultivar which would perform well in all environments.

However, an analysis of the genotype and environment interaction performed in this study identified groups of trial locations with similar or identical rankings of cultivars. Such groups of trial locations create mega-environments [25,26], i.e., a group of environments in which the same the same cultivars are yielding best. Based on the assessment of 36 cultivars in 11 trial locations, four mega-environments were identified. The use of GGE biplots helped to indicate these mega-environments and the cultivars that yield the best in each of them. When a cultivar's best performance is limited to one or several mega-environments, the cultivar is characterized by a narrow adaptation. A cultivar with a narrow adaptation should be recommended for a specific mega-environment.

A recommendation based on the correct assessment of cultivars can be very useful for farmers. It is important for the METs system to have appropriate trial locations where the sugar beet yield will be tested [27]. These locations can represent the target region for sugar beet cultivation, i.e., a mega-environment. Unfortunately, the locations designated by the GGE biplot into one mega-environment for sugar beet cultivars are not adjacent to each other and do not form a coherent geographic region. It is also impossible to identify other causes or factors (e.g., soil properties or climate) that combine individual locations in designated groups of environments (mega-environments). Despite attempts to evaluate the relationship between these locations—i.e., a comparison of soil pH, NPK contents, soil textures, and rainfall or average daily temperatures—no compounds were found. There is no biological reason for the division into these specific mega-environments. Similarly, as in the case of the sugar beet [28,29], the lack of justification for the division of locations into a mega-environment was observed in many other species of arable crops [30,31].

The results presented here show that it is worth applying the mega-environment approach and, consequently, recommending sugar beet cultivars for cultivation in specific regions (mega-environments or target regions). Even in individual mega-environments, there might be a location with different soil and/or climatic conditions. An example would be group 4, which has the most fertile soils and favorable climate conditions. Locations belonging to this group are scattered all over the country. This is very often the case when a mega-environment is created based on the evaluation of cultivars adaptation patterns (a posteriori) [32] by just using the yield data from METs and using statistical analysis methods such as GGE or AMMI (Additive Main Effects and Multiplicative Interaction). This is different from the a priori grouping of environments into mega-environments based on their similar climatic and soil properties. [33]. An example may be the designated agro–ecological regions for wheat in Poland [22] and Pakistan [12].

The creation of mega-environments was criticized in the past [34] as unnecessary due to similar environmental conditions in European countries with a temperate climate (e.g., Czech Republic, Germany and Poland). It was assumed that cultivars' rankings would be similar (with this same superior cultivars). However, in the case of our analysis, we showed that the ranking of cultivars was very different in individual locations. As different superior cultivars were indicated in different locations, we can group locations with the same best-yielding cultivars and similar cultivar rankings.

This information can be useful even afterwards to validate cultivar recommendations, increase their efficiency, and increase their effectiveness. It makes sense to create such mega-environments despite the lack of unambiguous characteristics combining the locations belonging to them.

The evaluation of sugar beet cultivar adaptation and the determination of mega-environments, carried out with the use of GGE biplots, was confirmed by the Spearman rank correlation test for the white sugar yield rankings of the tested cultivars between the trial locations. A consistency between the results of the evaluation of cultivars' rankings and the GGE biplots was also observed for other crop species, including wheat [35,36]. The Spearman rank correlation analysis showed a compatibility of the cultivars' white sugar yield rankings between the examined years. This means that the same cultivars were considered as superior, regardless of the year. It can be concluded that various weather conditions in years did not change the white sugar yield ranking of sugar beet cultivars. The cultivars considered were characterized by a high stability of white sugar yield in each growing season. The high compatibility of the white sugar yield rankings between the years can contribute to a more effective recommendation of cultivars. Such compatibility increases the chances of accurate cultivar recommendation to farmers. In conclusion, this research broadens the knowledge of the adaptation patterns of sugar beet cultivars. Assigning trial locations to mega-environments will allow a more effective cultivar recommendation for growers.

In summary, based on the analysis of the genotype–environment interaction, it was not possible to indicate one cultivar or cultivars with a wide adaptation to the environmental conditions of the temperate climate. However, we have indicated cultivars with a narrow adaptation to locations grouped in mega-environments. On the basis of the same or very similar rankings, we could designate four groups of environments (mega-environments). The results of cultivar evaluation are reproducible, which suggests a small variability of white sugar yield under the influence of weather conditions in these three growing seasons. The applied methodological and statistical approach allowed for an effective recommendation of sugar beet cultivars for cultivation in temperate climate environments.

**Author Contributions:** Conceptualization, M.S. and Z.W.; methodology, M.S. and T.L.; formal analysis, M.S. and K.N.; data curation, T.L.; writing—original draft preparation, M.S.; writing—review and editing, M.S. and E.W.-G.

**Conflicts of Interest:** The authors declare no conflict of interests.

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
