# Peer review of "Assessment of Stability and Adaptation Patterns of White Sugar Yield from Sugar Beet Cultivars in Temperate Climate Environments"

_agronomy, doi:10.3390/agronomy9070405_

Round 1
Reviewer 1 Report
This is a nice paper detailing sugar yield over 3 years at 11 locations with 36 cultivars.
The manuscript needs improvement in readability (check correct use of tense, plurals, and other minor editing).
Overall, there is a lack of integration in the stated purpose of the study, and the conclusions drawn from the study. For instance, the definition of meg-environments is an interesting one, however, salient characteristics for each mega-environment are only marginally referenced through the use of a sucrose yield 'bioassay' (e.g. cultivar trial). What other features define a mega-environment? Are they contiguous?
Specific points:
Line 43: Beets produce roots, not tubers.
Line 65: How does one evaluate 36 cultivars over 3 consecutive seasons if only 20 cultivars were studied each growing season?
Table 1: An accepted soil classifier should also be presented (e.g. http://www.fao.org/soils-portal/soil-survey/soil-classification/world-reference-base/ or similar).
Tables 2 and 4: What do Min and Max refer to? How is it important?
Line 112: Is 0.5% of total yield variation significant?
Figure 1: It is not at all clear how four groups (mega-environments) are extracted from this information. Perhaps a table of cultivars ranked according to location would help. I would have expected to be able to rank cultivars by some stability index across all locations. How well would such a stability index serve as a predictor of overall cultivar performance?
Figure 2: Black text on dark blue background is hard to read.
Author Response
Dear Reviewer,
Thank you for your comments concerning our manuscript entitled "Assessment of stability and adaptation patterns of white sugar yield from sugar beet cultivars in temperate climate environments ". Those comments are all valuable and very helpful for revising and improving our paper. We have studied comments carefully and have made correction which we hope meet with approval. Below we present our answers to reviewer comments
This is a nice paper detailing sugar yield over 3 years at 11 locations with 36 cultivars.
The manuscript needs improvement in readability (check correct use of tense, plurals, and other minor editing).
Response: The manuscript was editing according to this comment
Overall, there is a lack of integration in the stated purpose of the study, and the conclusions drawn from the study. For instance, the definition of meg-environments is an interesting one, however, salient characteristics for each mega-environment are only marginally referenced through the use of a sucrose yield 'bioassay' (e.g. cultivar trial). What other features define a mega-environment? Are they contiguous?
Response: Unfortunately, the designated mega-environments do not form a geographically uniform area, and you cannot identify the factors that connect the locations. This effect is often found for different species of crops. Especially if mega-environments have been designated based on the yield results of the cultivars observed in many growing seasons (mega–environments a posteriori). We have modified the Discussion sections to better present and justify this problems. We also gave in this section the possible use of mega-environments created in this way.
Specific points:
Line 43: Beets produce roots, not tubers.
Response: It was changes
Line 65: How does one evaluate 36 cultivars over 3 consecutive seasons if only 20 cultivars were studied each growing season?
Response: Very often in the METs experience system, data sets are incomplete. The linear mixed models used in our work with the REML method are standard used for incomplete data. The use of incomplete data and its analysis by the REML method is widely accepted and effective for many species of arable crop
Table 1: An accepted soil classifier should also be presented (e.g. http://www.fao.org/soils-portal/soil-survey/soil-classification/world-reference-base/ or similar).
Response: We add to table 1 soil classification according to IUSS Working Group WRB. World Reference Base for Soil Resources 2014. International Soil Classification System for Naming Soils and Creating Legends for Soil Maps. As Reviewer 2 suggested
Tables 2 and 4: What do Min and Max refer to? How is it important?
Response: It was changes
Line 112: Is 0.5% of total yield variation significant?
Response: We added to this table evaluation of the variance components significant.
Figure 1: It is not at all clear how four groups (mega-environments) are extracted from this information. Perhaps a table of cultivars ranked according to location would help. I would have expected to be able to rank cultivars by some stability index across all locations. How well would such a stability index serve as a predictor of overall cultivar performance?
Response: We used one of the basic tools used to evaluation of cultivars adaptation patterns - GGE biplot. The interpretation of this type of figure has already been described many times. We acknowledge that the use of a different approach (eg. Table, only stability measure) will be ineffective and certainly even more difficult to interpretation. The stability measures have a disadvantage, they do not take into account the value of the cultivars yield. The analysis using GGE biplot simultaneously takes into account the stability of the cultivars as well as the values of their yield.
Figure 2: Black text on dark blue background is hard to read.
Response: We have increased all the figures at manuscript, which increases their readability. Unfortunately, changing the colors of numbers to white, makes the plot is unreadable against a bright blue and bright red background
Reviewer 2 Report
Line 5, correct Lenartowicz
Line 14, not true, fertilization? sowing date? etc.
Line 51, correct sugar beet
Line 74, please using IUSS Working Group WRB. World Reference Base for Soil Resources 2014. International Soil Classification System for Naming Soils and Creating Legends for Soil Maps. Update 2015;World Soil Resources Raport 106; FAO: Rome, Italy, 2015; 188p
Line 75 please explain whether it is a biological sugar yield or pure sugar yield?
Similarly line 87, 100, 101 and 102, 103, 105, 110, 116, 130, 152, 163, 168, 176, 205, 207, 208, 209
Line 75 correct [t ha-1]
Line 94 correct [9]
Line 108 correct [13].
Line 111, 112 Yield what?
Line 130-137 - correct t ha-1
Line 138 – t ha-1
Line 171-213 – very short discussion
Line 213 correct growing.
Line 213 No conclusions
Line 219-222 – only 22 references
Author Response
Dear Reviewer,
Thank you for your comments concerning our manuscript entitled "Assessment of stability and adaptation patterns of white sugar yield from sugar beet cultivars in temperate climate environments ". Those comments are all valuable and very helpful for revising and improving our paper. We have studied comments carefully and have made correction which we hope meet with approval. Below we present our answers to reviewer comments
Line 5, correct Lenartowicz
Response: It was done
Line 14, not true, fertilization? sowing date? etc.
Response: It was changes
Line 51, correct sugar beet
Response: It was done
Line 74, please using IUSS Working Group WRB. World Reference Base for Soil Resources 2014. International Soil Classification System for Naming Soils and Creating Legends for Soil Maps. Update 2015;World Soil Resources Raport 106; FAO: Rome, Italy, 2015; 188p
Response: We add to table 1 soil classification according above reference.
Line 75 please explain whether it is a biological sugar yield or pure sugar yield?
Response: In our manuscript we considered observation for white sugar yield (pure sugar yield). We added to Material and Methods Sections description about procedure used to calculated sugar yield in each trials.
Similarly line 87, 100, 101 and 102, 103, 105, 110, 116, 130, 152, 163, 168, 176, 205, 207, 208, 209
Response: See above.
Line 75 correct [t ha-1]
Response: It was done
Line 94 correct [9]
Response: It was done
Line 108 correct [13].
Response: It was done
Line 111, 112 Yield what?
Response: It was changes
Line 130-137 - correct t ha-1
Response: It was done
Line 138 – t ha-1
Response: It was changes
Line 171-213 – very short discussion
Response: We significantly increased Discussion section
Line 213 correct growing.
Response: It was changes
Line 213 No conclusions
Response:We added to Discussion section paragraph with conclusions and summary
Line 219-222 – only 22 references
Response: We added references